# Sol–Gel Synthesis and Characterization of a Quaternary Bioglass for Bone Regeneration and Tissue Engineering

**DOI:** 10.3390/ma14164515

**Published:** 2021-08-11

**Authors:** Ricardo Bento, Anuraag Gaddam, José M. F. Ferreira

**Affiliations:** 1CICECO—Aveiro Institute of Materials, Department of Materials and Ceramic Engineering, University of Aveiro, Santiago University Campus, 3810-193 Aveiro, Portugal; ricardobento@ua.pt (R.B.); anuraagg@ua.pt (A.G.); 2Instituto de Física de São Carlos, Universidade de São Paulo, São Carlos 13566-590, SP, Brazil

**Keywords:** bioactive glasses, alkali-free, sol–gel, bone regeneration, tissue engineering

## Abstract

Sol–gel synthesis using inorganic and/or organic precursors that undergo hydrolysis and condensation at room temperature is a very attractive and less energetic method for preparing bioactive glass (BG) compositions, as an alternative to the melt-quenching process. When properly conducted, sol–gel synthesis might result in amorphous structures, with all of the components intimately mixed at the atomic scale. Moreover, developing new and better performing materials for bone tissue engineering is a growing concern, as the aging of the world’s population leads to lower bone density and osteoporosis. This work describes the sol–gel synthesis of a novel quaternary silicate-based BG with the composition 60 SiO_2_–34 CaO–4 MgO–2 P_2_O_5_ (mol%), which was prepared using acidified distilled water as a single solvent. By controlling the kinetics of the hydrolysis and condensation steps, an amorphous glass structure could be obtained. The XRD results of samples calcined within the temperature range of 600–900 °C demonstrated that the amorphous nature was maintained until 800 °C, followed by partial crystallization at 900 °C. The specific surface area—an important factor in osteoconduction—was also evaluated over different temperatures, ranging from 160.6 ± 0.8 m^2^/g at 600 °C to 2.2 ± 0.1 m^2^/g at 900 °C, accompanied by consistent changes in average pore size and pore size distribution. The immersion of the BG particles in simulated body fluid (SBF) led to the formation of an extensive apatite layer on its surface. These overall results indicate that the proposed material is very promising for biomedical applications in bone regeneration and tissue engineering.

## 1. Introduction

Noticeable increases in life expectancy have been achieved over the past two centuries, as well documented in several literature reports [1,2]. The overall scientific progress made in the medical field, especially over the most recent decades, has greatly contributed to an increase in the population’s quality of life and life expectancy. Such progresses together brought an unavoidable increase in the average age of the population, with social, economic, and medical consequences [3,4]. Older people are less capable of physical exertion, and the reduction of the mechanical stimuli of the bones, along with the degradation of the tissues due to a lifetime of use, increases the risk of fractures [5,6]. The consequent increased incidence of bone-related diseases (e.g., osteoporosis, trauma fractures, removal of tumours, etc.) challenges researchers to provide new therapeutic solutions to prevent the onset of osteopathy, and to develop bone grafts to treat these ailments. Autografts, although still considered the gold standard due to their optimal osteogenic, osteoinductive, and osteoconductive properties, have multiple drawbacks, including the donor site morbidity and their limited availability [7]. On the other hand, allografts are often associated with risk of infection and a high non-union rate with host tissue [8,9]. Therefore, tissue engineering—and bone tissue engineering in particular—presents itself as a most promising alternative solution to the current bone grafting approaches [10].

Several materials have been tried, and have achieved some success. Among these materials, bioceramics and bioglasses are promising candidates for bone regeneration, owing to their natural properties—such as biocompatibility, osteoinduction, and osteoconduction—adding to their composition, which is similar to that of bone [11]. Dense ceramics, such as alumina [12,13] or zirconia [14,15,16], have found a wide use in load-bearing applications due to their excellent mechanical properties and low in vivo toxicity. However, the paradigm changed with the discovery of bioactive glasses (BGs) by Larry Hench et al., due to their ability to bond to living tissues through the formation of an interfacial bone-like hydroxyapatite layer when the bioglass is put in contact with biological fluids in vivo. Among a number of tested compositions, the one exhibiting the highest bioactivity index became well known, and has been trademarked as 45S5 Bioglass^®^ since 1985 [17]. Since then, the topic has received increasing attention, inspiring many other investigations aimed at further exploring the in vitro and in vivo performances of this BG, or gradually developing other related BG compositions [18,19,20]. Even though these glasses possess excellent affinity with native bone, their commercial success has so far been relatively limited. Their drawbacks include inadequate mechanical properties and high dissolution rates of alkaline ions in biological conditions, which may have a harmful impact on cells [21,22,23,24,25,26], along with several other shortcomings, as well summarized elsewhere [27]. Conversely, it has been demonstrated that well-designed alkali-free bioactive glass compositions offer a great number of advantages, including the ability to achieve full densification before the onset of crystallization, a fast biomineralization capability, with the formation of a crystalline surface apatite layer after immersion in SBF solution for 1 h [28,29], and good in vivo performances [30].

Traditionally, bioactive glasses are produced by melt-quenching. However, this technique incurs higher energy costs, and often results in partial devitrification with the formation of crystalline phases, which have reduced biological properties and are unsuitable for producing porous scaffolds [11]. Due to this, the sol–gel method has been regarded as a favourable alternative, as it yields glasses with higher purity, requiring low densification temperatures and allowing the production of a varied amount of compositions [31,32]. This technique does, however, exhibit some drawbacks—namely, the high cost of the precursors, and the difficulty in producing dense, monolithic pieces [33]. The present work aims at using water as a single solvent in the sol–gel synthesis of a novel, well-balanced, and thermally stable alkali-free BG composition intended for biomedical applications in bone regeneration and tissue engineering.

## 2. Materials and Methods

### 2.1. Bioglass Synthesis

A quaternary bioactive glass with a composition consisting of 60 SiO_2_–34 CaO–4 MgO–2 P_2_O_5_ (mol%) was prepared via the sol–gel method, following a detailed procedure described elsewhere [32]. Briefly, tetraethyl orthosilicate (TEOS, Si(O_2_H_5_), ≥98%), supplied by Sigma-Aldrich, and triethyl phosphate (TEP, O_4_P(C_2_H_5_O), ≥98%), supplied by Merck Schuchardt, were used as Si and P network precursors, respectively, while calcium nitrate tetrahydrate (Ca(NO_3_)_2_·4 H_2_O), supplied by Panreac, and magnesium nitrate hexahydrate (Mg(NO_3_)_2_·6 H_2_O), supplied by Scharlab, were selected as Ca and Mg network modifiers, respectively. Nitric acid (HNO_3_ ≥ 65%) supplied by Labkem was used as a catalyst to promote the hydrolysis of network precursors. Each preparation was planned to yield 0.2 mol of bioactive glass by mixing 1.46 g of TEP, 25.00 g of TEOS, 2.05 g of magnesium nitrate, and 16.06 g of calcium nitrate. Two separate aqueous solutions were initially prepared: one containing the network precursors, and the other the network modifiers. In brief, the required amounts of TEOS and TEP were added together to 20 mL of deionized water acidified with two drops of concentrated nitric acid, under magnetic stirring for 30 min, until obtaining a transparent sol. The solution of the network modifiers was prepared in parallel by simply adding the required amounts Ca(NO_3_)_2_·4 H_2_O and Mg(NO_3_)_2_·6 H_2_O to 20 mL of deionized water under magnetic stirring for 30 min. Due to their high solubility, this solution soon became transparent. Afterwards, both solutions were mixed together and magnetically stirred for a further 60 min, before being poured into Petri dishes and stored in an oven for 24 h at 100 °C to promote a relatively rapid sol–gel transition and drying. After 24 h, the xerogel was crushed with an agate pestle and mortar into a fine powder, and then heat treated at different temperatures.

### 2.2. Thermal Treatment

Ground xerogel powder was calcined for 2 h at temperatures of 600, 700, 800, and 900 °C, with a heating rate of 0.5 °C min^−^^1^, and the powders’ properties were assessed through several characterization techniques, as described below.

#### 2.2.1. XRD Characterization

The powders’ crystalline phase content was determined by X-ray diffraction (XRD, Rigaku Geigerflex D/Mac, C Series, Tokyo, Japan) using Cu K_α_ radiation with 2θ varying from 5–70° in steps of 0.026 s^−1^.

#### 2.2.2. FTIR Characterization

The functional groups of the samples were analysed by FTIR (FTIR Bruker Tensor 27) over the wavenumber range of 2000–300 cm^−1^, with 256 scans and 4 cm^−1^ resolution.

#### 2.2.3. SEM Imaging

Specimens were covered with a thin (15-nm) carbon layer using a thin film deposition system (PVD 75, Kurt J. Lesker Co., Jefferson Hills, PA, USA) before being examined using a Hitachi S4100 scanning electron microscope with a 15.0-kV accelerating voltage.

#### 2.2.4. Specific Surface Area Analysis

Specific surface area (SAA) and pore analyses were carried out with the BET, BJH, and t-plot methods (Micrometric Gemini M-2380); N_2_ was used as an adsorbate, and samples were previously degassed at 200 °C.

#### 2.2.5. SBF Immersion Assays

Powder samples were immersed in SBF for a period of 28 days. Every 7 days, the sample’s reaction with the SBF was halted by filtering, rinsing with deionized water, and drying.

## 3. Results and Discussion

### 3.1. Bioglass Synthesis—Effect of Mixing Time

The XRD patterns of the bioglasses prepared according to previous works [32] displayed an unexpected crystalline phase upon calcining at 600 °C. Since the synthesis experiments were carried out in the winter, an explanatory hypothesis for this was the relatively low room temperatures (15–17 °C) prevailing in the lab, which might have delayed the reaction kinetics. Therefore, the duration of the mixing step was increased from 30 to 60 min in order to allow complete hydrolysis. This longer mixing period successfully produced amorphous glasses, as shown in Figure 1.

### 3.2. Effect of Heat Treatment Temperature on the Relevant Properties of Bioactive Glass Samples

Heat treating the sol–gel-derived bioactive glasses is important to burn off the organics and release the nitrate ions from the Ca and Mg precursors. Increasing the heat treatment temperature enhances the densification of the material, affecting its overall physical properties.

#### 3.2.1. Crystalline Phase Assemblage

Heat treatment beyond a certain temperature level is likely to promote devitrification. Therefore, the ideal heat treatment temperature for the sintering step can be assessed by XRD. Figure 2 shows that the samples calcined within the temperature range 600–800 °C remain amorphous.

However, relatively well-defined crystalline peaks appear in the XRD pattern of the sample heat-treated at 900 °C. The crystalline phases were identified as silicon oxide (SiO_2_), calcium silicate (Ca_2_SiO_4_), calcium phosphate (Ca_3_(PO_4_)_2_), and calcium magnesium nitrate (Ca_3_Mg(SiO_4_)_2_). These results suggest that if an amorphous material is envisaged, the calcination temperature should not go much beyond 800 °C.

#### 3.2.2. Specific Surface Area

The impact of calcination temperature on the specific surface area (SSA) of the powders was assessed by resorting to gas adsorption of N_2_, using the BET technique. The evolution of SSA as a function of temperature is shown in Figure 3.

It can be seen that the SSA after calcination at 600 °C is still relatively high (~160 m^2^ g^−1^), and gradually decreases (to ~130 m^2^ g^−1^) with temperature increasing to 700 °C. This is followed by a drastic change (to ~16 m^2^ g^−1^) with a further increase in the calcination temperature to 800 °C, and then by a more gradual decrease to ~2–3 m^2^ g^−1^ upon heat treatment at 900 °C.

#### 3.2.3. Adsorption and Desorption Isotherms

The porous structure profiles of the samples were initially investigated by analysing their respective adsorption and desorption isotherms. These plots are shown in Figure 4. After an initial intersection between adsorption and desorption curves at relatively low pressures, associated with the formation of the N_2_ monolayer, a hysteresis loop can be observed due to the condensation of the N_2_ at higher pressures within the finer pores, creating multilayers. Such curves are classified as type IV according to IUPAC. At 900 °C, the curves do not intersect, producing an isothermal profile without a classification in IUPAC. It is known that the finer pores are the first to be eliminated due to their high driving force for densification. Accordingly, the average pore size tends to increase with an increase in the heat treatment temperature. On the other hand, some pores might have shrunk into the microporous scale and become less accessible (more impervious) to gas exchanges. This explains why condensation is less evident for the sample calcined at 900 °C, rendering the BET method ineffective at these ranges, and producing an isotherm with an apparently open loop.

Regarding the different hysteresis loops found in these type IV isotherms, at 600 and 700 °C, an H5 loop—characterized by a slight delay in desorption before a descending curve—can be observed, pointing towards a structure where both open and partially blocked mesopores are present. For the powder sample calcined at 800 °C, an H2 (Figure 4c) hysteresis can be observed, indicated by the larger delay before desorption, with a steeper, quasilinear, descending curve. This loop also correlates to pore blocking, but with a greater range of pore neck widths.

#### 3.2.4. Pore Size Distribution

The pore distribution profiles obtained through the BJH method describe the surface area of these powders in even greater detail. The contributions of the different pore sizes to the overall surface area (dA/dlog(w)) and pore volume (dV/dlog(w)) are displayed in Figure 5 and Figure 6, respectively. For all of the powders, the majority of the contribution to the total area and pore volume came from pores within the 18–200 Å range. The data depict the aforementioned decrease in total surface area and volume; however, the same data also depict an interesting trend regarding the porous evolution of these samples. Comparing the samples calcined at 600 °C with those calcined at 700 °C, the decrease in surface area is paired with a shift of the relative contribution to the total area from the smaller pores to larger ones. From 700 °C to 900 °C, this trend is reversed, and we observe that the major contribution to the total values comes from increasingly smaller pore ranges, highlighting the transition from a mesoporous to a microporous material. This shift is likely derived from the closure of the smaller pores in the 600–700 °C transition, which means that the larger pores, which are still open, are the main responsible factor for the measured surface area. With further increases in temperature, these pores will continue shrinking, resulting in smaller pores in the 800 and 900 °C samples.

### 3.3. Bioactivity Assessment through SBF Immersion Assays

The bioactivity of the bioactive glass powders was assessed by immersion in simulated body fluid (SBF) over a period of 28 days. Considering the generally high SSA values of the samples, the amount of powder that was added per mL of the SBF solution was double compared to the standard ratio of 0.5 cm^2^ mL^−1^ proposed by Popa et al. [34] for bulk samples, calculated according to the following equation:(1)$$SM=\frac{\rho \cdot SA\cdot D}{6}$$
where *SM* is the sample mass in grams, *ρ* is the density in grams per cubic centimetre, *SA* is the sample area in square centimetres exposed to the SBF solution, and *D* is the average particle diameter in centimetres. The tested powders were assumed to be perfect spheres with a diameter of ~1.25 × 10^−2^ cm (~125 µm, the average aperture size of the two sieves used, 150 μm and 100 μm). For 20 mL of SBF, 0.06 g of bioglass powder was added. Samples were retrieved every 7 days, and the reaction halted, and were then analysed via XRD and FTIR measurements, as well as through scanning electron microscopy.

#### 3.3.1. XRD Patterns

The XRD patterns for the SBF-treated glass powders can be seen in Figure 7. Calcium phosphate and calcite were the two phases formed on the surface of the particles. Peaks at 24 and 32° were present for all tested timepoints, and can be associated with apatite reflections. Other peaks in the regions around 29, 40, 47, and 50° can be attributed to apatite reflections as well. These crystalline peaks indicate the presence of a hydroxyapatite layer on the powders’ surface. Both crystalline phases formed are bioactive and prone to binding between the implant material and the living tissues.

#### 3.3.2. FTIR Spectra

The FTIR spectra further point towards the formation of a hydroxyapatite layer, as seen in Figure 8a. The spectra reveal the three main vibrational modes of the Si–O–Si groups—namely, the peaks in the 1000–1100 cm^−1^ range are associated with the stretching vibration of the Si–O–Si groups, with the bridging oxygens moving opposite of the silicone; at ~780 cm^−1^, a peak associated with the bending vibration of the Si–O–Si group, characterized by the oxygen atom moving at right angles; and finally, at 500 cm^−1^, the rocking vibration of the Si–O–Si is observed. Phosphate groups can also be identified by the peak at 600 cm^−1^ assigned to asymmetric bending. The peak at the ~1500–1550 cm^−1^ range is associated with a C–O stretch, indicating that carbon from the atmosphere may have reacted with calcium from the powders to form carbonates. Another carbonate stretching vibration can be observed at ~1300 cm^−1^_,_ related to unhydrolysed residual organics from the sol–gel step. Finally, the band located at ~1600 cm^−1^ is attributed to the hydrogen bonded to water that was absorbed in the sample [35,36,37]. To better understand the formation of hydroxyapatite due to SBF, the spectra were subtracted from the base glass, corresponding to 0 days, as seen in Figure 8b. The subtracted spectra reveal peaks at 1384, 1147, 968, 854, 744, 669, and 518 cm^−1^_,_ which correspond to the following vibrational modes: asymmetric stretching of CO_3_^2−^, asymmetric stretching of PO_4_^3−^, symmetric stretching of PO_4_^3−^, asymmetric bending of CO_3_^2−^, symmetric bending of CO_3_^2−^, stretching of PO_4_^3−^, and bending of PO_4_^3−^, respectively [37]. The spectra show that with the increasing number of days, the amount hydroxyapatite also increases.

#### 3.3.3. SEM Images

SEM observations further sustain the previous data by showing a relatively smooth surface for the control powder samples (0 days, calcined at 600 °C), whereas after 28 days, a rugged apatite surface with small circular apatite crystals can be observed (Figure 9).

## 4. Conclusions

This work reports on the fabrication and processing of a new bioactive glass belonging to the SiO_2_–CaO–MgO–P_2_O_5_ system via the sol–gel method. The effects of calcination temperature on the crystalline phase assemblage and on the specific surface area of these powders was studied. The results show an accentuated decreasing trend in the surface area with increasing temperature. At temperatures >> 800 °C, the samples undergo crystallization. According to BET results, the sample calcined at 600 °C shows the highest surface area, thus being suitable for biological applications, including the storage and release of pharmaceutical drugs. This sample also shows the formation of a hydroxyapatite layer after only 7 days of SBF treatment, and is therefore a promising material for future applications in bone regeneration and tissue engineering.

## Figures and Tables

**Figure 1 materials-14-04515-f001:**
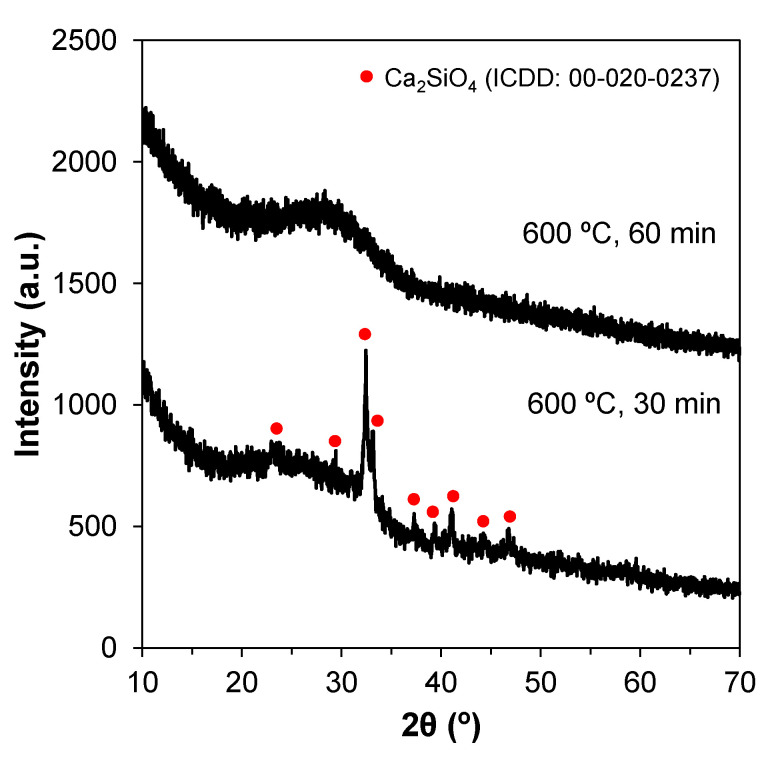
XRD patterns showing the effect of mixing times on the production of amorphous glasses via the sol–gel method. Changing the mixing period from 30 to 60 min produced amorphous glasses.

**Figure 2 materials-14-04515-f002:**
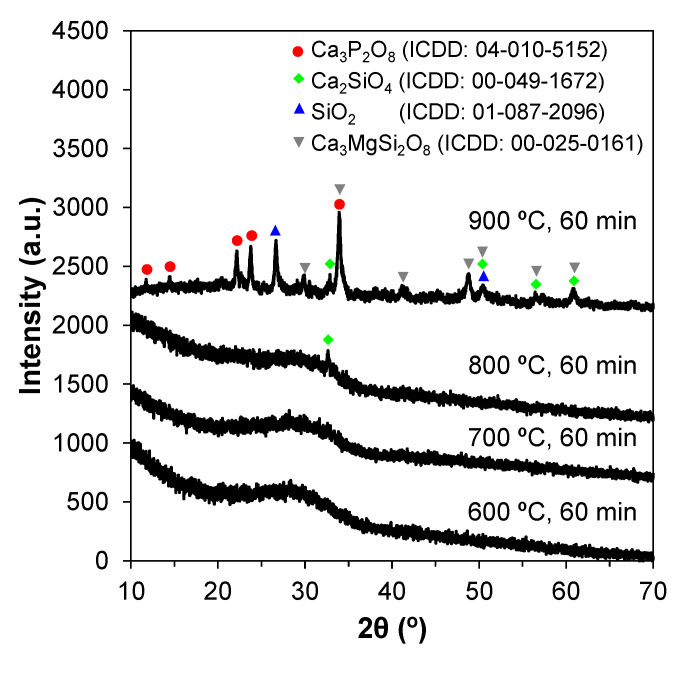
Effect of calcination temperature on glasses produced via the sol–gel method. Crystalline phases appear at 900 °C.

**Figure 3 materials-14-04515-f003:**
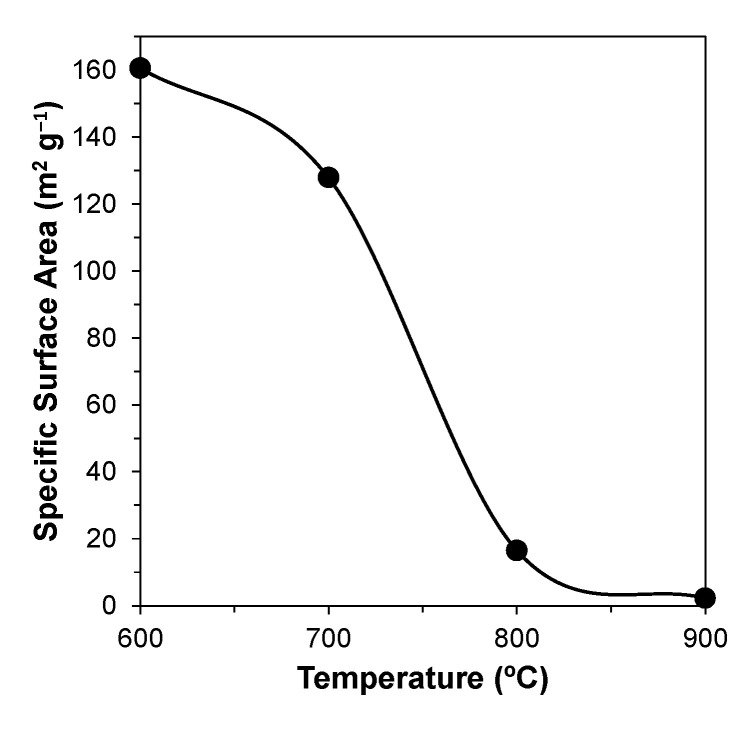
Effect of temperature on the specific surface area of the powders.

**Figure 4 materials-14-04515-f004:**
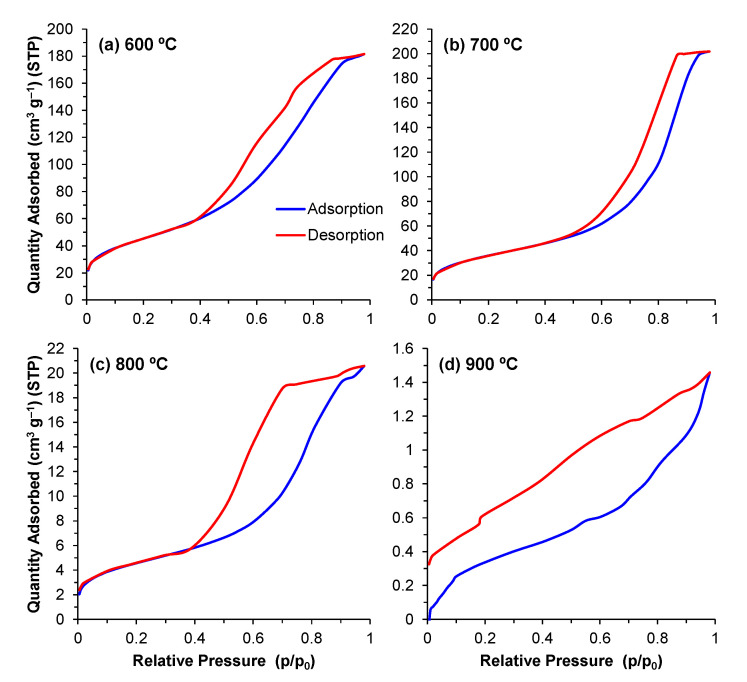
(**a**–**d**) Isotherm curves for powders calcined at temperatures of 600–900 °C.

**Figure 5 materials-14-04515-f005:**
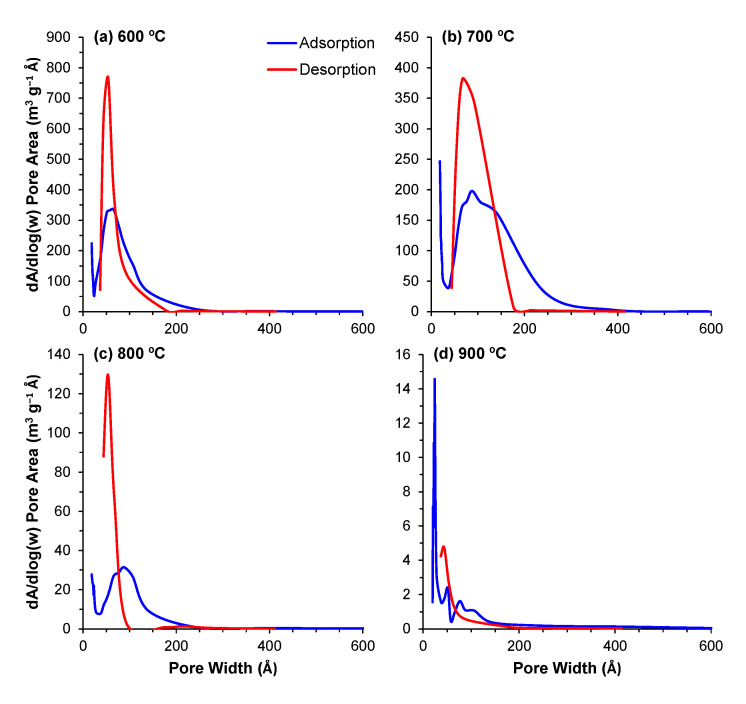
(**a**–**d**) Contribution of the different pore sizes to the overall surface area for the powders calcined at temperatures of 600–900 °C.

**Figure 6 materials-14-04515-f006:**
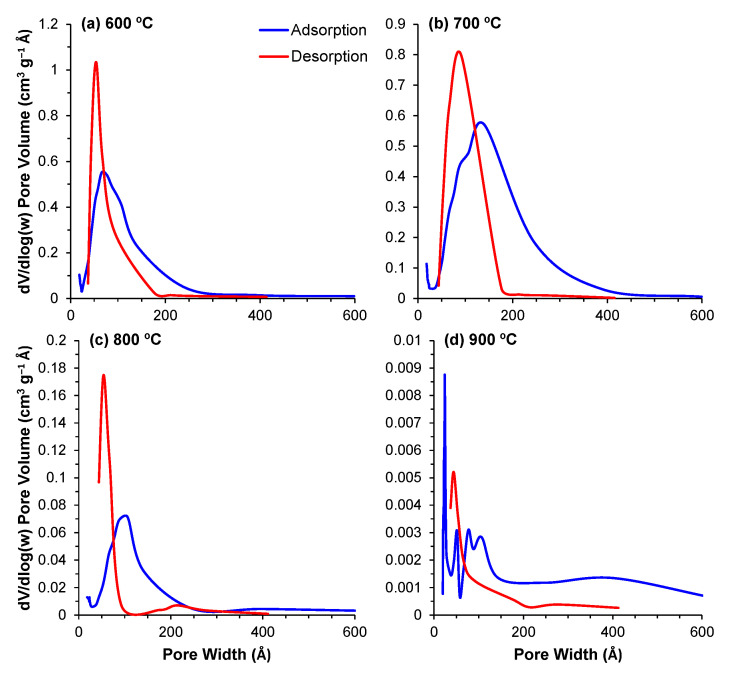
(**a**–**d**) Contribution of the different pore sizes to the overall pore volume for the powders calcined at temperatures of 600–900 °C.

**Figure 7 materials-14-04515-f007:**
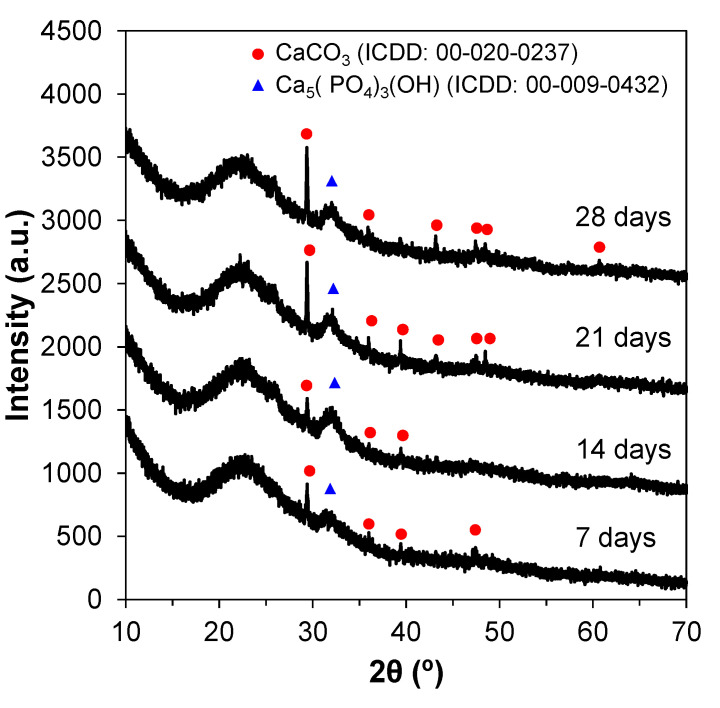
XRD patterns of powders (calcined at 600 °C) immersed in SBF from 7 to 28 days.

**Figure 8 materials-14-04515-f008:**
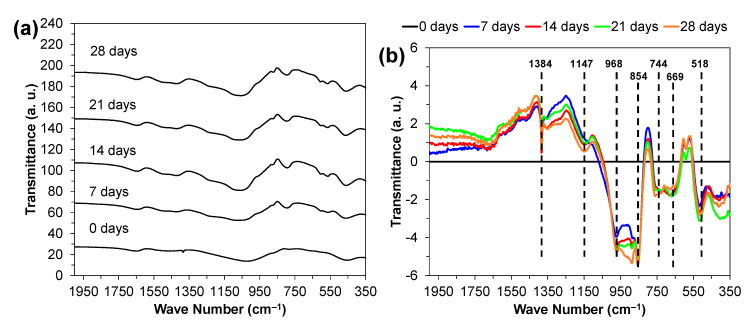
(**a**) FTIR spectra for powders that were calcined at 600 °C before immersion in SBF for different durations; (**b**) FTIR spectra with the 0-day sample baseline subtracted.

**Figure 9 materials-14-04515-f009:**
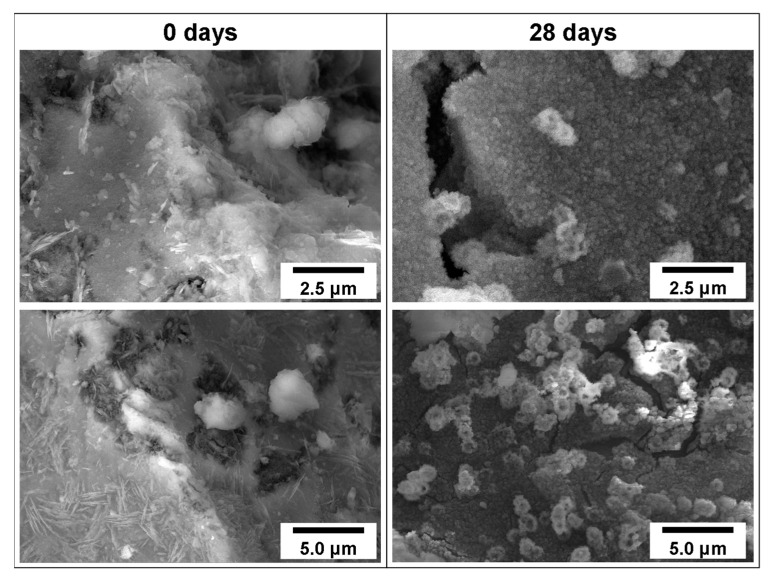
SEM images of powder (calcined at 600 °C) surfaces before (0 days) and after immersion in SBF for 28 days, confirming the formation of a carbonated hydroxyapatite layer.

## Data Availability

Not applicable.

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
