# Peer review of "Sol–Gel Synthesis and Characterization of a Quaternary Bioglass for Bone Regeneration and Tissue Engineering"

_materials, 2021, doi:10.3390/ma14164515_

Round 1
Reviewer 1 Report
Although interesting and important in its topic, the manuscript suffers from many shortcomings.
Title contains 'for bone regeneration and tissue engineering', however, there is no research on this performed.
Abstract should be concise. The presented one is extremely long! Most of the abstract (first two thirds) would be more appropriate in the introduction section.
Introduction lacks the aim of the work. The article is about synthesis and characterization of a novel (?) bioglass. Most of the space contains information which could be condensed into a couple of sentences (paragraph about life expectancy and increased incidence of bone diseases).
The title evokes that the novelity of the manusript is in the way of synthesis of a new promising material, Materials and Methods section states 'a detailed procedure is described elsewhere', 'In brief,...'
Stating 'Since we were in the winter season' does not sound scientifically. Especially in such a type of article, conditions should be well defined due to reproducibility.
Figure 9 description: appetite ?
Author Response
The authors are thankful for the critical comments received from the worthy Reviewer, which helped us to increase the clarity and the quality of the manuscript. Below please find our answers, point by point, to the comments received.
- Title contains 'for bone regeneration and tissue engineering', however, there is no research on this performed.
Answer: The Reviewer is right; the manuscript does not include any experimental data related to the appraisal of bone regeneration and tissue engineering capabilities. The expression is just intended to convey to the readers the applications intended for the sol-gel derived bioactive glass. This is an ongoing post-graduation research work and the aim is to evaluate in a near future the biological performance of the material through either in vitro and in vivo tests.
- Abstract should be concise. The presented one is extremely long! Most of the abstract (first two thirds) would be more appropriate in the introduction section.
Answer: Thanks for the pertinent suggestion, which was fully addressed by the authors by making the required changes in the Abstract and in the Introduction parts of the revised manuscript.
- Introduction lacks the aim of the work. The article is about synthesis and characterization of a novel (?) bioglass. Most of the space contains information which could be condensed into a couple of sentences (paragraph about life expectancy and increased incidence of bone diseases).
Answer: Thanks for the useful suggestions. Accordingly, the Introduction has been improved by moving and integrating some parts of the Abstract, condensing some information related to life expectancy and increased incidence of bone diseases phrases, and by clearly stating the aim of the work.
- The title evokes that the novelity of the manusript is in the way of synthesis of a new promising material, Materials and Methods section states 'a detailed procedure is described elsewhere', 'In brief,...'
Answer: Thanks for pointing out to this apparent incongruency. Please note that the novelty of the manuscript is not only in the way the promising material is synthesized by sol-gel using acidified water as a single solvent, but also a novel and better-balanced alkali-free composition. Most of the works dealing with sol-gel synthesis use co-solvents do help dissolving the alkoxide precursors, and chelating agents to stabilise the most reactive metal ions to prevent segregation of the components. As far as we know, only a few works from our research group have reported the use of acidified water as a single solvent in sol-gel synthesis, justifying the expression “as described elsewhere”.
- Stating 'Since we were in the winter season' does not sound scientifically. Especially in such a type of article, conditions should be well defined due to reproducibility.
Answer: Thanks for the remark. The room temperature conditions (15-17ºC) in the lab in winter season are indicated in the revised manuscript.
- Figure 9 description: appetite ?
Answer: Thanks for the remark. The typo was corrected in the revised manuscript.
Reviewer 2 Report
In its current form the manuscript is very interesting for the community of researchers working with bioglass. Beyond minor editing issues the paper provides a complete description of the effect of temperature on bioglass preparation. One issue that I feel would further strengthen the manuscript is the presentation of the XRD and FTIR spectra that are solely relate to samples calcinated at 600C. I feel it would benefit the audience to see the same results for the samples treated at 700 and 800C as they have the potential to reveal interesting trends of great value. In the intro also Older people appears as Holder people (please check typos).
Author Response
Thanks for the comments provided. The mentioned typos has been corrected in the revised manuscript.
Please also note that the data plotted in Figure 1 is for samples calcined at 600 ºC, which have been mixed for different times (30, and 60 minutes) to show the relevancy of allowing a sufficient mixing/hydrolysis time during the sol-gel synthesis. After that, there is no reasoning for further exploring the samples obtained under the shorted hydrolysis time. Therefore, Figure 2 only presents the XRD data of samples corresponding to the longer hydrolysis time (1 hour) and calcined at 600 ºC, 700 ºC, 800 ºC and 900 ºC.
Concerning the FTIR spectra, the authors regret to say that during the experimental work carried out in the frame of a Master Thesis, data were collected solely for the sample calcined at 600 ºC as it was the one selected to further explore the material for the fabrication of porous scaffolds by Robocasting. Another manuscript related to this last part of the work is under preparation. But for the XRD and FTIR spectra reported in this manuscript will be referred to in that manuscript under preparation.
Reviewer 3 Report
This a scientifically sound study reporting the manufacturing by sol-gel and characterization of a phosphate-containing bioglass. The characterization of the material is very thorough from the chemical-physical point, but further experiments regarding the biological activity of the glass (e.g. cytotoxicity, osteogenesis...) are not presented. The manuscript is clear and well written.
Albeit quite limited in scope, I think the paper could be suitable of publication on Materials with some minor cosmetic corrections as in the file attached.

Author Response
This a scientifically sound study reporting the manufacturing by sol-gel and characterization of a phosphate-containing bioglass. The characterization of the material is very thorough from the chemical-physical point, but further experiments regarding the biological activity of the glass (e.g. cytotoxicity, osteogenesis...) are not presented. The manuscript is clear and well written.
Albeit quite limited in scope, I think the paper could be suitable of publication on Materials with some minor cosmetic corrections as in the file attached.
Answer: The authors are thankful to the worthy Reviewer for the favourable comments and recommendation provided. The manuscript was carefully revised considering all the useful suggestions received.
Round 2
Reviewer 1 Report
The authors have substantially improved the manuscript and have addressed my suggestions. I have no further points.
One typo in the introduction: 'Holder people' Older people ?
Author Response
The authors are thankful to the reviewer for recommending our manuscript for publication in its present form.